# Increased Expression of Superoxide Dismutase 2 Is an Indicator of Worse Prognosis of Oropharyngeal Cancer

**DOI:** 10.3390/ijms26073223

**Published:** 2025-03-30

**Authors:** Yoshitaka Aoki, Satoru Kondo, Eiji Kobayashi, Makiko Moriyama-Kita, Hirotomo Dochi, Shigetaka Komura, Yosuke Nakanishi, Kazuhira Endo, Naohiro Wakisaka, Tomokazu Yoshizaki

**Affiliations:** 1Division of Otolaryngology and Head and Neck Surgery, Graduate School of Medical Science, Kanazawa University, Kanazawa 920-8640, Ishikawa, Japan; ltd.express.0755@gmail.com (Y.A.); e_kobayashi@med.kanazawa-u.ac.jp (E.K.); mkita@med.kanazawa-u.ac.jp (M.M.-K.); h_dochi@med.kanazawa-u.ac.jp (H.D.); komushige@med.kanazawa-u.ac.jp (S.K.); nakanish@med.kanazawa-u.ac.jp (Y.N.); endok@med.kanazawa-u.ac.jp (K.E.); wakisaka@med.kanazawa-u.ac.jp (N.W.); tomoy@med.kanazawa-u.ac.jp (T.Y.); 2Department of Otorhinolaryngology, NHO Kanazawa Medical Center, Kanazawa 920-8650, Ishikawa, Japan

**Keywords:** SOD2, oropharyngeal cancer, HPV

## Abstract

Human papillomavirus (HPV) is a known risk factor for oropharyngeal cancer (OPC), with distinct HPV-positive and HPV-negative subtypes. Reactive oxygen species have been implicated in the carcinogenesis of several malignancies. Superoxide dismutase 2 (SOD2), a mitochondrial enzyme, is highly influenced by oxidative stress. This study investigated whether SOD1 and SOD2 expression in OPC affects primary tumor progression, lymph node metastasis, stage, and overall survival (OS). Biopsy or surgically resected specimens from 72 patients with OPC were analyzed via immunohistochemical staining for SOD1 and SOD2. The proportion of stained cells within the tumor area was assessed. Associations between SOD1 and SOD2 expression, T classification, N classification, and stage were evaluated. Factors correlated with OS in OPC were also examined. No significant differences in SOD1 or SOD2 expression were observed concerning T or N classification, or stage. However, high SOD2 expression was identified as a significant poor prognostic factor for OS. Regardless of HPV status, SOD2 expression predicts poor prognosis in OPC. Evaluating SOD2 expression may help predict OPC prognosis.

## 1. Introduction

Smoking and alcohol consumption are the primary risk factors for oropharyngeal cancer (OPC). However, recent studies have shown that human papillomavirus (HPV) is also involved in the development of more than half of OPC cases [1,2]. HPV-mediated OPC is typically characterized by positive p16 staining. In cases of p16-positive OPC, HPV-deoxyribonucleic acid (DNA) is more frequently detected compared with other upper aero digestive tract cancers or non-cancer cases [1,2]. As HPV-positive OPC is more sensitive to radiation and chemotherapy than HPV-negative OPC, it generally has a better prognosis [3]. The 8th edition of the Union for International Cancer Control (UICC) Tumor–node–metastasis (TNM) classification, published in 2017, differentiates OPC staging based on p16 positivity, indicating HPV-association [3,4,5].

Reactive oxygen species (ROS) that are involved in cancer development have been identified. Mitochondria actively generate ROS through the electron transport chain and oxidative phosphorylation [6]. Increased ROS levels induce intracellular oxidative stress, leading to DNA damage and a higher likelihood of cancer development [6,7]. ROS are also considered to play an important role in the development of head and neck cancer [8]. Mitochondria is a major site of ROS production. Superoxide dismutase (SOD) is the primary enzyme responsible for breaking down ROS generated during cellular metabolism. Copper/zinc superoxide dismutase (Cu/Zn-SOD, SOD1) is found in the cytoplasm, whereas manganese superoxide dismutase (Mn-SOD, SOD2) is localized in the mitochondria [9,10,11]. In the research on SOD1, several studies have reported its association with cancer. SOD1 plays a role in tumor growth in melanoma and non-small cell lung cancer [12,13]. Another report shows that SOD1 activity may be essential for maintaining cellular ROS below a critical threshold [14]. Many studies on SOD2 have also reported an association with cancer. Reduced SOD2 expression levels might lead to ROS accumulation, which can increase the risk of cancer development. Studies on hepatocellular carcinoma have reported that decreased SOD2 expression correlates with a shorter survival [15]. In contrast, high expression levels of SOD2 are associated with a worse prognosis in patients with breast cancer [16]. A previous study showed that SOD2 is strongly expressed in cervical intraepithelial neoplasia grade 3 and squamous cell carcinoma compared with non-neoplastic tissues [17]. On the other hand, there is a report that HPV oncogenes such as E7 affect SOD2 activity using oral cell and hypopharyngeal cancer cell [18,19].

Whether SOD1 or SOD2 expression levels are related to OPC remain unknown. In this study, we aimed to investigate whether SOD1 and SOD2 expression levels in OPC tissues influence disease progression, including primary tumor progression, lymph node metastasis, and survival prognosis.

## 2. Results

### 2.1. Clinicopathological Analysis of SOD1 and SOD2 Expression Levels in OPC Tissues

Among 72 patients diagnosed with OPC between January 2007 and December 2018, SOD1 expression was positive in 66 cases (91.7%) and negative in 6 cases (8.3%), whereas SOD2 expression was positive in 47 cases (65.3%) and negative in 25 cases (34.7%) (Figure 1). No significant difference was observed in all clinical factors between positive and negative SOD1 cases (Table 1). On the other hand, a significant difference was observed in the selection of the initial treatment between positive and negative SOD2 cases, and no significant differences were observed in other clinical factors between positive and negative SOD2 cases (Table 2).

### 2.2. Immunohistochemical Analysis of SOD1 and SOD2 Expression Levels in OPC Tissues

When we examined SOD1 expression in T classification (*p* = 0.107, Figure 2A), N classification (*p* = 0.600; Figure 2B), and stage (*p* = 0.201, Figure 2C), no significant differences were observed in any category.

When we examined SOD2 expression in T classification *(p* = 0.271, Figure 3A), N classification (*p* = 0.091; Figure 3B), and stage (*p* = 0.620; Figure 3C), no significant differences were observed in any category.

### 2.3. Association Between the Expression and Overall Survival (OS)

We analyzed the relationship between SOD1 or SOD2 expression and prognosis in 72 patients with OPC. The five-year OS rate was 80.1%. The five-year OS rate was 70.4% for patients with high SOD1 expression, whereas it was 66.7% for those with low SOD1 expression, showing no significant difference (Figure 4). On the other hand, the five-year OS rate was 60.0% for patients with high SOD2 expression, whereas it was 90.2% for those with low SOD2 expression, showing a significant difference (Figure 5). We showed that the high SOD2 expression group had a significantly shorter survival than the low expression group. Next, we assessed the impact of 12 factors, including sex, age, subsite, T classification, N classification, stage, initial treatment (surgery or chemoradiotherapy), p16 expression, SOD1 expression, SOD2 expression, smoking history, and alcohol consumption, on OS using Cox regression analysis. Finally, SOD2 expression and stage were found to significantly influence OS. OPC cases with SOD2 expression had a significantly shorter OS than those without SOD2 expression (*p* = 0.024, hazard ratio [HR] 4.359, confidence interval [CI] 1.210–15.699). Additionally, the advanced stage had a significantly shorter OS than the early stage (*p* < 0.001, HR 1.826, CI 1.298–2.569). No significant differences were observed in other factors related to OS (Table 3).

## 3. Discussion

SOD2 plays a role in ROS detoxification. However, whether SOD2 suppresses carcinogenesis remains unclear. Enhancing SOD2 activity, which reduces oxidative stress, may prevent DNA damage and hinder cancer development [20,21,22,23]. In contrast, SOD2 activity can neutralize ROS, potentially preventing cell necrosis and increasing the likelihood of malignant transformation. Studies on gastric and esophageal cancers have shown that SOD2 activity is stronger in these cancers than in the normal mucosa [24]. Additionally, invasive breast cancer cell lines exhibit higher SOD2 activity than non-invasive cell lines [16,25].

The prognostic significance of SOD2 expression remains controversial. Some studies have associated low SOD2 activity with poor prognosis [15,20], whereas others have suggested that higher SOD2 activity is associated with worse outcomes in gastric [24], esophageal [26], breast [16,25], lung [27,28,29], cervical [17], and penile [30] cancers.

This study demonstrated that SOD2 expression is a prognostic factor for OS in OPC, independent of primary tumor size or lymph node metastasis. These findings suggest that mitochondrial antioxidant mechanisms may represent the distinct carcinogenesis pathways of OPC [17].

Current OPC staging and treatment are based on the UICC 8th edition, which assumes no prognosis differences based on HPV status. This study suggests that prognosis can be predicted based on the presence or absence of SOD2 in addition to the presence or absence of HPV.

The reason for the worse prognosis in cases with high SOD2 activity may be increased ROS activity, which promotes DNA damage and carcinogenesis. However, this increase in ROS detoxification via enhanced SOD2 activity could be compensatory, reflecting an attempt to neutralize increased ROS levels.

In radiation therapy, the depletion of SOD2 can lead to increased radiosensitivity [31,32]. SOD2 expression can be used to be a biomarker determine whether radiation therapy is effective or not [33].

Several factors influence SOD2 activity, including hexokinase 2 (HK-2) activation and heavy smoking, which has been linked to increased SOD genes in oral cells [18,34,35].

SOD2 has been identified as a potential target gene of Homeobox A13 (HOXA13), a key factor in esophageal carcinogenesis [26]. On the other hand, mitochondrial DNA displacement loop (D-loop) instability is more pronounced in lymph node metastases than in primary tumors, further supporting the role of mitochondrial factors in colon cancer progression [36]. Further research is necessary to elucidate the mechanisms underlying the increase in SOD2 expression and therapeutic implication.

In cervical cancer, genetic studies have explored the correlation between HPV integration patterns and SOD2 gene polymorphisms [37,38]. Moreover, SOD2 is strongly expressed in cervical intraepithelial neoplasia grade 3 and squamous cell carcinoma compared with non-neoplastic tissues [17]. In addition, HPV oncogenes such as E7 affect SOD2 activity [18,19]. These reports suggest HPV infection associates SOD2 expression. However, SOD2 expression influences OS independent of p16 expression which indicates HPV infection in OPC. This discrepancy may be due to tissue tropism.

SOD1 is also involved in ROS detoxification, but its role in carcinogenesis remains unclear. Some studies have shown that increased SOD1 levels are associated with increasing tumor activity in the lung [13] and breast [39]. Some proteins and microRNAs that may be involved in SOD1 expression have also been reported [40]. However, only a few reports that predict OS about SOD1 have shown this [41]. In this study, SOD1 was positive at a high rate at the time of initial diagnosis, suggesting the possibility that SOD1 may be involved in the onset of OPC. Conversely, the few cases in which SOD1 was negative made it difficult to predict the prognosis. This study has suggested that SOD2 is more sensitive than SOD1 in predicting prognosis for OPC.

This study has some limitations. First, the number of patients who underwent chemoradiotherapy was small. The surface area of tissue samples from patients who underwent surgery was considerably larger, even for T1 lesions, than that of biopsy specimens from patients who received chemoradiotherapy, which may have increased the reliability of the staining proportion. As biopsy specimens represent only a small portion of the tumor’s surface, they may not fully reflect the entire primary tumor. Second, although we established a relationship between SOD2 expression and prognosis, we did not account for factors that may influence the SOD2 activity, such as HK-2 and HOXA13. Third, we did not assess the prognosis of patients who received chemoradiotherapy based on their SOD2 expression levels, as only a small number of patients opted for this treatment during the study period. Furthermore, because this was a retrospective study, treatment selection was not based on SOD2 expression. However, further studies are mandatory to address these limitations.

## 4. Materials and Methods

### 4.1. Patients

This study included 72 patients with OPC who visited Kanazawa University Hospital between 2007 and 2018. Tissue specimens were obtained and verified through pathological sectioning, with patient consent.

Clinical data, including the presence or absence of p16, TNM stage, and treatment method, were collected. TNM staging followed the 8th edition of the UICC classification [5]. Distant metastasis was identified via radiological examination. OS, which was defined as the time from diagnosis to death, date of censoring owing to loss to follow-up, or last follow-up, was used as a measure of prognosis and ranged from 1 to 121 months. The Ethics Committee of Kanazawa University approved this study (Institutional Review Board #2012-031).

### 4.2. Immunohistochemistry

The specimens were obtained from patients who underwent surgery for the primary lesion (surgical specimens) or those who received non-surgical treatments (biopsy specimens).

Samples were fixed in 10% formaldehyde and embedded in paraffin. Sections were deparaffinized in Hemo-De (Falma, Tokyo, Japan) and rehydrated in an ethanol gradient. Endogenous peroxidase activity was blocked with methanol containing 3% hydrogen peroxide for 10 min after washing three times with phosphate-buffered saline (PBS) at a pH = 7.2. The slides were boiled in 10 mM sodium citrate retrieval buffer, adjusted to a pH = 6 at 95 °C for 15 min for antigen retrieval. After cooling to 25 °C, followed by rinsing with PBS, the tissue sections were incubated with a protein block (Dako, Glostrup, Denmark) for 20 min and incubated overnight at 4 °C with primary antibodies: mouse anti-SOD1 (1:50, Santa Cruz Biotech., Dallas, TX, USA) or mouse anti-SOD2 (1:50, Santa Cruz Biotech., Dallas, TX, USA).

The sections were then washed three times with PBS. The sections were incubated with EnVision + secondary antibody (Dako) for 30 min. The reaction products were developed by immersing the sections in 3′,3-diaminobenzidine tetrahydrochloride solution. Subsequently, the sections were counterstained with hematoxylin (Fujifilm, Tokyo, Japan).

### 4.3. Evaluation of SOD1 and SOD2 Immunoreactivity

Two investigators (Y.A. and S.K.), blinded to the clinical data, independently evaluated the stained sections using an IX83 microscope (Olympus, Tokyo, Japan). Entire sections were scanned at medium (×100) and high (×400) magnification. Immunoreactivity was scored based on the proportion and intensity of staining.

The percentages of immunoreactive cells demonstrated a wide range of staining expression (from 0% to 100%) and one peak for SOD1 and SOD2 expression demonstrated a frequency < 10%. On the basis of these data, the cases were classified into negative and positive categories, as follows: negative, <10% immunoreactive cells; positive, ≥10% immunoreactive cells.

### 4.4. Statistical Analysis

Statistical analyses were performed using the IBM SPSS Statistics version 29 (IBM, New York, NY, USA). Clinical characteristics of the patients were analyzed using chi-square or Fisher’s exact test. Staining intensities were compared using the Kruskal–Wallis test or Tukey’s *t*-test. Prognostic factors were evaluated using Cox regression analysis incorporating 12 variables, including sex, age, subtype, T classification, N classification, stage, initial treatment (surgery or chemoradiotherapy), p16 expression, SOD1 expression, SOD2 expression, smoking status, and alcohol consumption. OS curves were estimated using the Kaplan–Meier method and calculated using the log-rank test. Statistical significance was set at *p* < 0.05.

## 5. Conclusions

This study is the first to demonstrate the utility of SOD2 expression in OPC. SOD2 expression can be an independent factor for predicting prognosis in OPC, regardless of p16 expression status. Evaluating SOD2 expression is useful for predicting the prognosis of OPC.

## Figures and Tables

**Figure 1 ijms-26-03223-f001:**
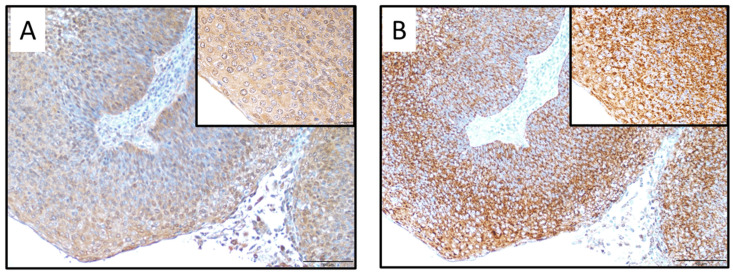
SOD1 or SOD2 immunostaining of specimens of OPC. (**A**) SOD1 expression was mainly observed in the cytoplasm of tumor cells (magnification ×100, scale bar = 100 μm). The upper right rectangle (magnification ×400, scale bar = 20 μm). (**B**) SOD2 expression was mainly observed in the cytoplasm of tumor cells (magnification ×100, scale bar = 100 μm). The upper right rectangle (magnification ×400 scale bar = 20 μm). (**C**) Low SOD1 expression. (**D**) Low SOD2 expression.

**Figure 2 ijms-26-03223-f002:**
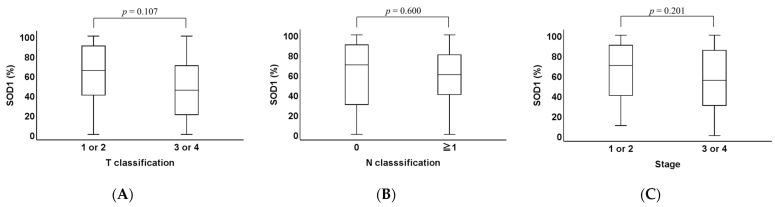
Comparison of SOD1 expression classified by T classification, N classification, and stage. (**A**) The SOD1 staining areas were compared across T classification. No significant differences were observed among the T categories (*p* = 0.107). (**B**) The SOD1 staining areas were compared with and without lymph node metastasis. No significant differences were found in the SOD1 staining area of OPC cases among the N classifications (*p* = 0.600). (**C**) The SOD1 staining areas were compared across disease stage. No significant difference was observed in the SOD1 staining area of the primary lesion among each stage (*p* = 0.201).

**Figure 3 ijms-26-03223-f003:**
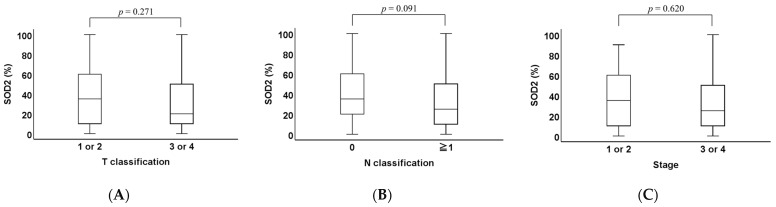
Comparison of SOD2 expression classified by T classification, N classification, and stage. (**A**) The SOD2 staining areas were compared across T classification categories. No significant differences were observed in SOD2 staining among T categories (*p* = 0.271). (**B**) The SOD2 staining areas were compared with and without lymph node metastasis. No significant differences were observed in the SOD2 staining among N classifications categories (*p* = 0.091). (**C**) The SOD2 staining areas were compared for each stage. No significant difference was observed in the SOD2 staining area of OPC cases among each stage (*p* = 0.620).

**Figure 4 ijms-26-03223-f004:**
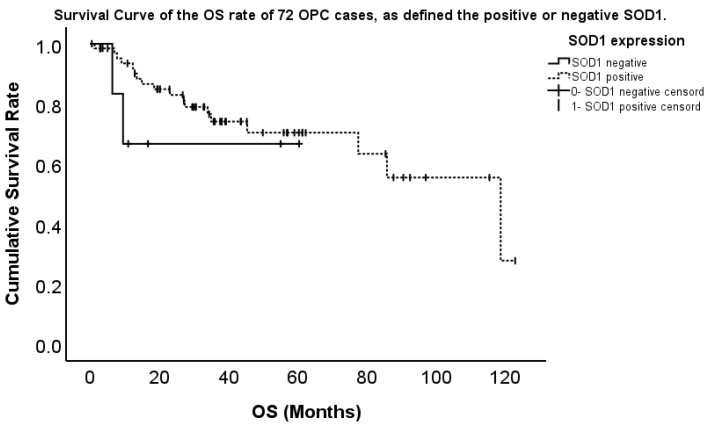
Survival curve of the OS of 72 OPC cases, as defined using the positive or negative SOD1.

**Figure 5 ijms-26-03223-f005:**
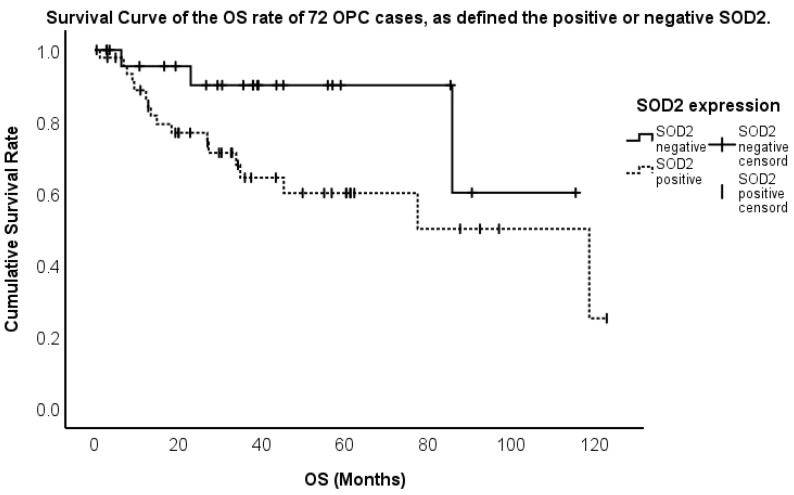
Survival curve of the OS of 72 OPC cases, as defined using the positive or negative SOD2.

**Table 1 ijms-26-03223-t001:** Patient background of 72 cases for OPC. Among these, 66 cases were SOD1 positive and 6 were SOD1 negative. Statistical analyses were performed by chi-square or Fisher’s exact test. No significant differences were observed in the other characteristics.

Characteristics	Total	SOD1-Positive	SOD1-Negative	*p* Value
Number of cases	72	66	6	
Sex				0.837
Men	62	57	5	
Women	10	9	1	
Age				0.431
Mean	63.5	64.9	52.8	
Range	33–89	43–89	33–74	
Medium	64	65	54	
T categories				1.000
T1–2	48	44	4	
T3–4	24	22	2	
N categories				0.459
N0	26	23	3	
N1–3	46	43	3	
Stage				0.560
1–2	44	41	3	
3–4	28	25	3	
Therapy				0.882
Surgery	46	42	4	
Chemoradiotherapy	26	24	2	
HPV				0.104
p16-positive	46	44	2	
p16-negative	26	22	4	
Smoking				0.793
Yes	57	52	5	
Never	15	14	1	
Alcohol				1.000
Yes	60	55	5	
Never	12	11	1	

**Table 2 ijms-26-03223-t002:** Patient background of 72 cases for OPC. Among these, 47 cases were SOD2 positive and 25 were SOD2 negative. Statistical analyses were performed by chi-square or Fisher’s exact test. Bold letter indicates significant difference (*p* < 0.05). A significant difference was observed in the selection of the initial treatment, whereas no significant differences were observed in the other characteristics.

Characteristics	Total	SOD2-Positive	SOD2-Negative	*p* Value
Number of cases	72	47	25	
Sex				0.274
Men	62	42	20	
Women	10	5	5	
Age				0.976
Mean	63.5	62.7	65.7	
Range	33–89	33–89	53–81	
Medium	64	63	66	
T categories				0.161
T1–2	48	34	14	
T3–4	24	13	11	
N categories				0.119
N0	26	20	6	
N1–3	46	27	19	
Stage				0.516
1–2	44	30	14	
3–4	28	17	11	
Therapy				**0.002**
Surgery	46	36	10	
Chemoradiotherapy	26	11	15	
HPV				0.395
p16-positive	46	36	10	
p16-negative	26	18	8	
Smoking				0.899
Yes	57	37	20	
Never	15	10	5	
Alcohol				0.580
Yes	60	40	20	
Never	12	7	5	

**Table 3 ijms-26-03223-t003:** Cox proportional hazard regression analysis of 72 patients with OPC and OS. Factors whose *p*-values were greater than 0.05 in the univariate models were not included (NI) in the multivariate analysis. Bold letter indicates significant difference (*p* < 0.05).

	Univariate		Multivariate	
Variates	Hazard Ratio (95% CI)	*p*	Hazard Ratio (95% CI)	*p*
Sex	0.847 (0.091–7.894)	0.884		NI
Age (>60)	1.576 (0.332–7.480)	0.567		NI
Subsite (tonsil vs. other subsites)	0.382 (0.103–1.412)	0.149		NI
T stage (T1, 2 vs. T3, 4)	1.036 (0.663–1.617)	0.877		NI
N stage (N0 vs. N1,2,3)	0.819 (0.221–3.039)	0.765		NI
Stage (I/II vs. III/IV)	**1.669 (1.032–2.699)**	**0.037**	**1.826 (1.298–2.569)**	**<0.001**
Surgery or Chemoradiotherapy	2.081 (0.476–9.094)	0.330		NI
p16-positive/negative	0.943 (0.278–3.201)	0.925		NI
SOD1-positive/negative	0.908 (0.142–5.802)	0.918		NI
SOD2-positive/negative	**8.422 (1.236–57.832)**	**0.030**	**3.915 (1.210–15.699)**	**0.024**
Smoking	1.644 (0.327–8.269)	0.546		NI
Alcohol	0.253 (0.043–1.490)	0.129		NI

## Data Availability

No other data were created or analyzed in this study.

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
