# Peer review of "Increased Expression of Superoxide Dismutase 2 Is an Indicator of Worse Prognosis of Oropharyngeal Cancer"

_ijms, 2025, doi:10.3390/ijms26073223_

Round 1

Reviewer 1 Report

Comments and Suggestions for Authors

The aim of the  study  was "to investigate whether SOD1 and SOD2 expression levels in OPC tissues influence disease progression, including primary tumor progression, lymph node metastasis, and survival prognosis." I found that authors in methods described SOD1 and  SOD2 thus in results I suspect to find all information related both. Authors focused in results more on SOD2, the same comment are to discussion.

example "Table 1. Patient background of 72 cases for OPC. Among these, 47 cases were SOD2 positive and 25 were SOD2 negative." add Table for SOD1 characteristic - even statistic are poor 66vs6.

The discussion must be extended with SOD1 informations

Regarding References: authors wrote: " Research on SOD1 suggests an association between SOD1 mutations and familial amyotrophic lateral sclerosis, although few studies have reported its association with cancer [12-14]"
Those  12-14 are  related to ALS, please  add those few studies with SOD-1 association with cancer. I found at least 10 references with main conclusion that "High expression of SOD-1, (...) are determinant of the larger tumor."

Reviewer 2 Report

Comments and Suggestions for Authors

Dear Authors,

Your manuscript "Increased Expression of Superoxide Dismutase 2 Is an Indicator of Worse Prognosis of Oropharyngeal Cancer," ijms-3525315, which examines the usefulness of SOD1 and SOD2 as diagnostic and prognostic markers of OPC, is written in an easy-to-follow way and contains some useful observations. Although the article lacks some originality, SOD1/2 has not been studied in OPC yet. Therefore, I recommend publishing your manuscript after making a few minor corrections:

  1. Statements in lines 33 – 37 need to be referenced.
  2. The tissue type should be added in the sentence in lines 58-59 and 162-163 (ref.18,19).
  3. Please, add the % in the results section (section 2.1., lines 66-68).
  4. All figures in the manuscript are missing titles. Please add them. For example, for Fig1, Immunohistochemical expression of SOD1 and SOD2 in OPC.
  5. The name of the applied statistical test should be added in Table 1, as well as the explanation of marking statistically significant results (e.g. bold, *, italic letter…).
  6. The text presented in section 2.2. should be shortened, as there is no statistically significant result.
  7. Please upscale the letters and numbers in the graphs (Fig 2 and 3).
  8. Please explain why only the expression of SOD2 was examined in section 2.3. Or, add the results for the association of SOD1 expression and OS of OPC patients.
  9. Evaluation of SOD1/2 immunoreactivity should be better described. Particularly, describe in detail what score 1 is, what score 2 is, and what was taken as a positive result (name the % of positive cells and the intensity).
  10. As there is no published article on SOD1/2 expression in OPC, I suggest you compare your results with TCGA data. Adding the TCGA analysis to the results section would significantly improve the quality of your article.

Round 2

Reviewer 1 Report

Comments and Suggestions for Authors

Authors answered all my doubts